# A Phase I/IIa Prospective, Randomized, Open-Label Study on the Safety and Efficacy of Nebulized Liposomal Amphotericin for Invasive Pulmonary Aspergillosis

**DOI:** 10.3390/jof10030191

**Published:** 2024-03-01

**Authors:** Jesús Fortún, Elia Gómez-García de la Pedrosa, Alberto Martínez-Lorca, Patricia Paredes, Pilar Martín-Dávila, Alicia Gómez-López, María José Buitrago, Javier López-Jiménez, Francesca Gioia, Rosa Escudero, Maria Elena Alvarez-Alvarez, Cruz Soriano, Javier Moreno-García, Diana San Miguel, Noelia Vicente, Santiago Moreno

**Affiliations:** 1Department of Infectious Diseases, Hospital Universitario “Ramon y Cajal”, Avda Colmenar Km 9,1, 28034 Madrid, Spain; pmartindav@gmail.com (P.M.-D.); francesca_gioia@hotmail.com (F.G.); rosa.escudero0@gmail.com (R.E.); alvarezalvarezmariaelena@gmail.com (M.E.A.-A.); javier.moreno@salud.madrid.org (J.M.-G.); smguillen@salud.madrid.org (S.M.); 2Instituto de Investigación Sanitaria Hospital “Ramón y Cajal” (IRYCIS), 28034 Madrid, Spain; eliagomez@gmail.com (E.G.-G.d.l.P.); albertoml85@yahoo.es (A.M.-L.); paredespatricia@gmail.com (P.P.); jljimenez@salud.madrid.org (J.L.-J.); cruzsorianocuesta@yahoo.es (C.S.); dianesms83@hotmail.com (D.S.M.); noelia.vicente@salud.madrid.org (N.V.); 3Department of Medicine, Universidad Alcalá, 28801 Madrid, Spain; 4Centro de Investigación Biomédica en Red de Enfermedades Infecciosas (CIBERINFEC), Instituto de 5 Salud Carlos III, 28029 Madrid, Spain; aliciagl@isciii.es (A.G.-L.); buitrago@isciii.es (M.J.B.); 5Department of Microbiology, Hospital Universitario “Ramon y Cajal”, 28034 Madrid, Spain; 6Department of Nuclear Medicine, Hospital Universitario “Ramon y Cajal”, 28034 Madrid, Spain; 7Mycology Reference and Research Laboratory, Centro Nacional de Microbiología, 28034 Madrid, Spain; 8Department of Hematology, Hospital Universitario “Ramon y Cajal”, 28034 Madrid, Spain; 9Intensive Care Unit, Hospital Universitario “Ramon y Cajal”, 28034 Madrid, Spain; 10Department of Pneumology, Hospital Universitario “Ramon y Cajal”, 28034 Madrid, Spain; 11Department of Pharmacy, Hospital Universitario “Ramon y Cajal”, 28034 Madrid, Spain

**Keywords:** invasive pulmonary aspergillosis, nebulized liposomal amphotericin B, ^18^F-fluorodeoxyglucose positron emission tomography

## Abstract

Although nebulized liposomal amphotericin B (NLAB) is being used in invasive pulmonary aspergillosis (IPA) prophylaxis, no clinical trial has shown its efficacy as a therapeutic strategy. NAIFI is the inaugural randomized, controlled clinical trial designed to examine the safety and effectiveness of NLAB (dosage: 25 mg in 6 mL, three times per week for 6 weeks) against a placebo, in the auxiliary treatment of IPA. Throughout the three-year clinical trial, thirteen patients (six NLAB, seven placebo) were included, with 61% being onco-hematological with less than 100 neutrophils/μL. There were no significant differences noted in their pre- and post-nebulization results of forced vital capacity (FVC), forced expiratory volume in 1 s (FEV1), and oxygen saturation between the groups. Neither bronchospasm nor serum amphotericin B levels were reported in any patients given NLAB. ^18^F-Fluorodeoxyglucose positron emission tomography (FDG-PET-TC) was carried out at the baseline and after 6 weeks. A notable decrease in median SUV (standardized uptake value) was observed in NLAB patients after 6 weeks (−3.6 vs. −0.95, *p*: 0.039, one tail). Furthermore, a reduction in serum substance galactomannan and beta-D-Glucan was identified within NLAB recipients. NLAB is well tolerated and safe for patients with IPA. Encouraging indirect efficacy data have been derived from image monitoring or biomarkers. However, further studies involving more patients are necessary.

## 1. Introduction

Despite advancements in treatment, invasive pulmonary aspergillosis (IPA) still results in high mortality rates. The most recent large-scale clinical trial compared the effects of two first-line drugs, isavuconazole and voriconazole, in patients with hemato-oncological IPA. The results showed a 20% mortality rate in both groups [1]. In larger case studies involving routine clinical practice, this mortality rate can rise to over 40% [2].

Liposomal, lipidic, and conventional nebulized amphotericin B are not currently approved for the treatment of IPA. However, various studies have examined their use as a preventative measure against fungal infections in lung transplant recipients [3,4,5], high-risk hematological patients [6,7], and patients with allergic broncho-pulmonary aspergillosis [8]. These studies have shown promising results, demonstrating high concentrations in the distal airway, no systemic toxicity, and good tolerance for the procedure. These factors suggest its potential as an adjunct treatment for IPA, specifically in broncho-pulmonary cases.

Experimental studies have verified the significant advantages of using nebulized amphotericin B for both treating and preventing IPA. A meta-analysis of eight studies examined 839 immunocompromised animals. The results indicated substantial decreases in mortality rates in those animals treated with nebulized amphotericin B compared with untreated animals. This effect was particularly notable when animals were treated prior to exposure to *Aspergillus*. No notable distinctions were found concerning efficacy and renal toxicity between standard and lipid amphotericin B. However, differences were found in pulmonary surfactant functionality when using conventional amphotericin B [9].

In humans, a recent comprehensive review, which included 13 case reports, 11 observational studies and 3 clinical trials, concluded that administering nebulized liposomal amphotericin B (NLAB) appears to be safe, with no severe side effects [10]. Our team recently detailed our experience using NLAB with 11 patients, which included adjunctive antifungal therapy in 5 patients and secondary prophylaxis in 6. After 3 months, we noted a marginally better clinical outcome in patients treated with NLAB. In a multivariate Cox regression analysis, accounting for any uncontrolled underlying disease revealed that NLAB use was linked to reduced mortality at the 12-month mark [11].

The ^18^F-fluorodeoxyglucose positron emission tomography FDG/PET/CT method offers an objective and quantifiable assessment of cellular metabolic activity, whether tumoral or infectious. This facilitates the monitoring of therapeutic effects, making it an outstanding surrogate marker for evaluating the therapeutic efficacy of IPA [12,13,14].

Thus far, the majority of the clinical use of NLAB has been centered on prophylaxis. No randomized controlled studies have been conducted to assess the therapeutic effectiveness of nebulized liposomal amphotericin B therapy.

The NAIFI study is a pilot, randomized phase I/IIa clinical trial that compares the safety and efficacy of NLAB (25 mg, three times per week) versus a placebo nebulization as an auxiliary therapy in patients having proved or probable IPA. The main goal is to assess the safety and tolerability of nebulized liposomal amphotericin B in these patients. Additional aims involve evaluating the clinical, radiological, and microbiological effectiveness of nebulized liposomal amphotericin B as a supplementary treatment for IPA.

## 2. Methods

### 2.1. Study Design and Inclusion Criteria

We included patients who were over 18 years old, diagnosed with IPA according to the recent EORTC/MSG criteria [15], and had provided informed consent. The diagnosis of probable IPA required either CT scan evidence showing dense, well-circumscribed lesions with or without a halo sign, air crescent sign cavity, or wedge-shaped consolidations, along with an isolation of *Aspergillus* species in respiratory samples or a positive serum or bronchoalveolar lavage (BAL) galactomannan (GM) test result, or a positive direct test.

For GM testing (both BAL and serum), we relied on the Ag Virclia^®^ Monotest (Vircell S.L., Granada, Spain, cut-off: 0.2). Fungal identification was determined by the microscopic examination of lactophenol cotton-blue stained slides and MALDI-TOF mass spectrometry (Bruker Daltonics, Bremen, Germany) following the manufacturer’s instructions.

All patients were treated with systemic antifungal therapy, including voriconazole, isavuconazole, and liposomal amphotericin B. We randomized patients for an open-label trial involving nebulized therapy. The interventions were nebulized liposomal amphotericin B (25 mg in 6 mL, three times per week for 6 weeks) for the study group or nebulized injection water (6 mL, three times per week, 6 weeks) for the control group. The systemic therapy duration was determined on an individual basis according to medical judgment.

### 2.2. Study Setting and Institutional Approvals

The prospective study was funded by the Spanish Ministry of Health’s ‘Fondo de Investigaciones Sanitarias’ (Health Research Fund; FIS grant, PI18/00179), and was conducted from October 2019 to October 2022 at a tertiary hospital in Madrid, Spain. The study’s promoter was Hospital Universitario Ramón y Cajal, and its protocol (version 3.0, 4 July 2019; NAIFI01 code, EUDRACT no: 2019-000745-12) was approved by the Institutional Review Board. Patient consent was duly obtained. Additionally, the Spanish Agency of Medicines and Sanitary Products (AEMPS) granted its approval on 13 August 2019; locator PXB29SAA6A.

### 2.3. Objectives

To ensure the safety of NLAB tolerance, forced vital capacity (FVC) and forced expiratory volume in 1 s (FEV1) readings were taken each week for the first 6 weeks. Any reductions greater than 20% in FVC and FEV1 after nebulization compared with pre-nebulization values were deemed significant. These readings were made using a micro-spirometer, courtesy of Micro Medical Ltd., based in Rochester, UK.

Additional measurements taken at each visit, pre- and post-nebulization, included oxygen saturation, blood pressure, symptoms such as cough and dyspnea, bronchospasm occurrence, need for bronchodilator treatment, and signs of nausea, vomiting, dysphagia, foul taste, and chest pain.

To support NLAB safety monitoring, serum amphotericin B levels were assessed using high-performance liquid chromatography coupled with ultra-violet detection (HPLC-UV). This involved a heat extraction and protein precipitation technique. For Amphotericin B identification and quantification, we used a photodiode array detector (PDA) to establish the molecule’s specific UV profile under set chromatographic conditions and to determine the peak absorption wavelength (405 nm).

To track the response in both arms, we conducted weekly tests during the first 6 weeks. These tests included the serum Ag Virclia^®^ Monotest (produced by Vircell S.L., Granada, Spain)., with a cut-off value of 0.2), the serum BDGlucan (made by Fujifilm Wako Chemicals Europe GmbH^®^, Neuss, Germany) with a cut-off of 7 pg/mL), and a customized serum PCR multiplex real-time PCR test to detect *A. fumigatus*, *A. terreus*, and *A. flavus*.

Furthermore, we utilized ^18^F-Fluorodeoxyglucose Positron Emission Tomography (FDG-PET-CT) at diagnosis and at the sixth week to monitor radiological responses. By measuring the standardized uptake value (SUV) index, which indicates the uptake of ^18^F-FDG in tissues relative to the administered dose, we could calculate metabolic activity.

The follow-up period ended at week 12. We examined several outcomes: (a) complete response, defined as both the resolution of signs and symptoms and over 90% radiological improvement in the CT scan; (b) partial response, characterized by more than a 50% radiological improvement; (c) stability, indicating a clinical response but less than 50% radiological improvement; and (d) progression or death.

### 2.4. Nebulization

eFlow^®^ (PARI Pharma Iberia, Madrid, Spain) nebulizers were used for nebulization [16].

### 2.5. Randomization

Patients were randomized 1:1 at baseline (day 0) using permuted block randomization. The randomization took into account onco-hematologic patients with severe neutropenia conditions (<100 cells/μL). This method ensured a homogeneous distribution across both groups.

### 2.6. Exclusion Criteria

The following exclusion criteria were applied: inability or refusal of the patient (or legal representative) to give consent; pregnancy or planning to become pregnant during the study; lactation; formal contraindication to administration of nebulized drugs; hypersensitivity to amphotericin B; confirmation at the time of diagnosis of extra-pulmonary aspergillosis; intubated patients or requiring imminent intubation at the time of randomization because some studies have confirmed that NLAB can precipitate in the breathing tubes; participation in another clinical trial in the previous month; and life expectancy < 1 week.

### 2.7. Concomitant Antifungal Therapy

To prevent bias, the study adhered to standard medical practices and international guidelines when administering systemic treatments to all patients. Therefore, any patient in the study might have received voriconazole, isavuconazole, or liposomal amphotericin B as systemic antifungal treatment. Any other antifungal treatment, if administered in the absence of those mentioned above, was deemed suboptimal.

Patients undergoing nebulized therapies with other antimicrobials, including antifungals, were not eligible for inclusion.

### 2.8. Compliance and Visits

The Pharmacy Department oversaw the tracking of treatment compliance and medication control for the study. In line with RD 1090/2015, they meticulously recorded the patient’s randomization number, supplied medication, batch number, and expiration date in the case report form (CRF).

The study consisted of nine visits: baseline visit (also known as visit 0 or screening visit), weeks 1–6, week 9, and lastly, week 12 (day 84, which entailed an overall evaluation and marked the conclusion of the follow-up). During the first 6 weeks, spirometric tests and the evaluation of factors relating to nebulization tolerance were conducted before and after nebulization at each visit. Outpatients were required to bring their nebulizers to each visit for on-site usage and monitoring.

### 2.9. Statistical Analysis

Standard descriptive statistics were used to summarize the study population characteristics. Categorical variables were compared with the χ^2^ test. The Student’s *t*-test or Mann–Whitney *U* test was applied for continuous variables. Repeated measurements across time points were compared using paired parametric or nonparametric tests (Student’s *t*-test for paired samples or Wilcoxon signed-rank test). The treatment of IPA naturally leads to an improvement in the radiological infiltrate and consequently to a reduction in the SUV index; therefore, for the statistical analysis, a one-tailed *p* was used to analyze the effect of nebulization with NLAB compared with the placebo. Continuous variables were previously dichotomized at the median values to be entered into the model. Statistical analysis was performed with SPSS version 20.0 (IBM Corp., Armonk, NY, USA).

## 3. Results

### 3.1. Characteristics of Patients

Over a 3-year period, 13 patients were enrolled (Table 1), including 6 who received NLAB and 7 who were given a placebo. Originally, the plan was to include 30 patients, but due to the SARS-CoV-2 pandemic and the subsequent prohibition of nebulized treatments at the facility, the study was halted for 20 months. Our subjects consisted of 6 males and 7 females, all of whom met the criteria for probable aspergillosis (EORTC 2019) [15].

Isolation of *Aspergillus* spp. in BAL registered positive in 61% of cases, with elevated GM in BAL and serum detected in 77% and 69% of cases, respectively. In two instances, fungal hyphae were also found in respiratory samples. *Aspergillus fumigatus* complex was confirmed in ten participants (77%), *Aspergillus terreus* in two, and *Aspergillus flavus* in one patient.

The majority of patients (61%) were undergoing onco-hematological treatments. All of them suffered from severe neutropenia (less than 100 WBC/µL). Two patients had autoimmune hepatitis and were being treated with high doses of corticosteroids. Another two patients had collagen diseases and were also undergoing corticosteroid and immunosuppressive therapy. One patient presented with alcoholism and malnutrition (Table 1).

### 3.2. Safety Analysis

The primary safety endpoint showed that NLAB was well tolerated. Nebulization was feasible in all patients, including those who received a placebo (nebulized water injection). Table 2 presents FEV1 values before and after nebulization, as well as the percentage change after administration. There were no significant differences in the decreases in FEV1 in patients receiving NLAB or the placebo. Only in three tests of the same patient with NLAB (#1) and in two tests of a placebo patient (#6) were reductions greater than 20% observed, which were predefined as significant. Moreover, the variance in FVC revealed no differences between the two groups, supporting the data observed in FEV1 (Table 3). No differences in oxygen saturation values were noted before or after nebulization.

Tolerance was similar in both groups; all patients experienced a cough during nebulization, but it was well tolerated. No instances of bronchospasm occurred, and three patients (two with NLAB, one placebo) needed prophylactic bronchodilator treatment before nebulization. One NLAB patient reported throat discomfort or irritation, and three reported a metallic taste during nebulization. There were no instances of shortness of breath, nausea, vomiting, or chest pain related to nebulization. Serum amphotericin B tests were performed during patients’ corresponding visits who were receiving NLAB by HPLC-UV. Notably, no serum amphotericin B levels were recorded in any test.

### 3.3. Clinical Response and Outcome

Table 1 presents data on patients’ clinical responses and outcomes. Patients #7 and #9 could not be evaluated due to their admission to the ICU at weeks +3 and +1, respectively. Nebulization was suspended, and their second FDG-PET-CT scans could not be conducted. Both patients exhibited progression in their underlying conditions. Of the remaining patients, seven displayed partial or complete responses at week 12 (four received NLAB, and three received a placebo). Four other patients did not demonstrate favorable clinical responses (two received a placebo and two received NLAB). Among these, three succumbed to the progression of their illnesses, independent of IPA evolution. Patient #11, who was on the placebo, exhibited a cavitated image and persistent microbiological presence at week 12. As a result, it was necessary to continue antifungal therapy. This patient also needed an increase in immunosuppressive treatment to manage her primary condition (systemic pulmonary sclerosis). Therapeutic drug monitoring was carried out in patients receiving voriconazole and the dose was adjusted to maintain serum levels between 1 and 5 ug/mL. No TDM was performed in patients receiving isavuconazole. Table 1 details the unique characteristics and progress of each patient.

### 3.4. Radiological Response, FDG-PET/CT Imaging, and Uptake Evaluations

In addition to the standard clinical evaluation at week 12, patients underwent a study using FDG-PET-CT at the start and at week 6. This method produces a quantitative measurement of the metabolic activity of targeted tissues through the SUV index. Figure 1 depicts the SUV indices for each patient’s pulmonary lesion, identified by nuclear medicine experts as suggestive of IPA, both at the start (week 0) and at week 6.

Due to technical constraints, the initial FDG-PET-CT was conducted between days 2 and 7, while the follow-up procedure at week 6 occurred between days 38 and 55. The time between these two tests for each patient is also shown in Figure 1. Overall, we observed a significant decrease in SUV at week 6 in patients treated with NLAB—a median SUV reduction of −3.6 compared with −0.95 in the placebo group [*p*: 0.039 (one-tailed); *p*: 0.052 (two-tailed)].

Except for patient #8, who started with a high initial SUV of 10.5, the beginning indices were comparable in both groups, with SUVs between 3.1 and 6.7. Notably, at week 6, except for patient #8 (who experienced a significant reduction to 4.7), four out of five patients treated with NLAB nebulization demonstrated an SUV of less than 2.3, compared with only one out of six in the placebo group. In fact, two placebo patients (patient #4 and patient #12) actually showed an increase in SUV at week 6 compared with their baseline readings.

### 3.5. Evolution of Serum GM and BDG and PCR

Figure 2 and Figure 3 display the GM and BDG results during the follow-up weeks, including the data from patients 7 and 9, who were not included in the clinical evaluation analysis. Despite the limited patient sample, Figure 2 exhibits a higher clearance rate among those who received NLAB. By week 3, five out of seven patients in the placebo group still had positive serum GM (>0.2, Virclia), compared with only two out of six patients in the NLAB group. The BDG results also imply a lower clearance rate in the placebo group, although the low specificity of this test constrains the reliability of this observation. The determination of PCR (multiplex real-time) in serum did not show positive results in any of the patients.

## 4. Discussion

This clinical trial shows that NLAB is safe and well tolerated in patients with IPA. Despite the small sample size, it provides data that suggest potential benefits when combined with standard systemic antifungal therapy. No prior randomized clinical trial has analyzed NLAB’s role in treating IPA, and no observational or controlled studies have evaluated its therapeutic efficacy.

In Hagiya’s [10] systematic review, 11 reports administered NLAB therapeutically, with six specifically targeting IPA [17,18,19,20,21,22]. All six reports indicated favorable outcomes. NLAB was also implemented against rare invasive mold infections and *Aspergillus* infection [23]. Out of 11 observational studies [3,4,5,12,24,25,26,27,28,29], only 1 was multi-centered; they all evaluated the prophylactic use of NLAB, primarily among lung transplant patients.

The majority of patients in the current study had angio-invasive forms of onco-hematological diseases. While nebulized therapies are typically more beneficial in broncho-pulmonary forms that affect mainly non-oncohematologic patients, the effectiveness of Nebulized Liposomal Amphotericin B (NLAB) in preventative care has been demonstrated in patients with hemato-oncological diseases. A rigorously designed, randomized controlled trial supported this, finding that NLAB preventative therapy significantly decreased the incidence of IPA in patients with prolonged neutropenia due to hemato-oncological diseases [6] without negatively impacting their pulmonary function [7]. The widespread use of NLAB as a preventative measure in lung transplant patients was also documented; a recent network meta-analysis suggested that NLAB might be the most effective preventative approach for lung transplant recipients [30].

In the current study, we administered a 25 mg dose three times a week. Earlier studies varied the NALB dosage from 15–25 mg twice daily to 20–50 mg two or three times weekly, frequently for prevention. Our goal was therapy, and we decided on the 25 mg thrice-weekly dosage based on Monforte et al.’s findings [3]. Despite the angio-invasive nature of aspergillosis in hematological patients, the importance of achieving high alveolar concentrations of amphotericin B in these patients is crucial for infection control since amphotericin B is not detected in the serum. Monforte et al. [3] demonstrated that amphotericin B concentration is high on the first day (>11 ug/mL) after the administration of a 25 mg dose of n-LAB and remains elevated one week later (>4 ug/mL) above the common MIC, suggesting that a concentration gradient may result on delivery of effective amounts. A smaller study also concluded that preventive NLAB does not affect the lipid content in pulmonary surfactants [25].

In our study, we used high-performance liquid chromatography with ultra-violet detection (HPLC-UV) to measure amphotericin B levels in the serum. We employed a heat extraction and protein precipitation method, which extracts the amphotericin B from the liposome and measures both liposomal amphotericin B and free amphotericin B concentrations in the serum. This confirmed the absence of serum amphotericin B levels in all patients who received NLAB at their corresponding visits.

Nebulizations were conducted using eFlow^®^ nebulizers, which are vibrating membrane types with ultrasonic frequency. They offer a high dispensed dose rate (63%) and a high flow rate (0.7 mL/min), significantly reducing the inhalation time from the conventional 10–15 min to just 3–5 min. Designed for home use, eFlow^®^ is an ideal choice for domestic applications.

In this study, no substantial decline was observed in FEV1 or FVC among patients. Noteworthy reductions, pre-established as significant if they exceeded 20%, were found only in three incidences involving the same NLAB patient and two incidences involving a patient on the placebo. Additionally, no significant differences in tolerance were observed between the two groups. Prophylactic bronchodilators were required by only three patients (two taking NLAB, one on placebo). In a separate clinical trial conducted using NLAB for prophylaxis, about half of the patients reported induced coughing and a displeasing taste. However, this trial lacked a placebo control group for comparison and to ascertain the true impact of amphotericin B nebulization. Still, there was a reduction in one-year survival rates among patients receiving NLAB prophylaxis [31].

Although long-term use of NLAB prophylaxis is generally tolerable, certain studies have observed a decrease in susceptibility to amphotericin B among both *Aspergillus* and non-*Aspergillus* species [5,26]. However, we do not consider this a risk when using NLAB therapeutically. This is due to the low resistance to amphotericin B and high concentrations within the alveolar region. In our study, we identified two isolates of *A. terreus* and one isolate of *A. flavus*. These are species with lower sensitivity to amphotericin B, but two resulted in favorable outcomes.

The limited sample size presents a challenge when evaluating clinical responses in these patients. The complete or partial response rates were 67% (four out of six) for the group treated with NLAB and 60% (three out of five) for the placebo group (*p*: ns). While this study does not definitively confirm a superior clinical response in patients receiving NLAB, some indirect data suggest an enhancement of response without observed toxicity, making NLAB a valuable option to be considered for further analysis in multicenter studies with a larger patient cohort. The incorporation of FDG-PET-CT scans at baseline and at week +6 allowed for an objective assessment of lung inflammation reduction in both groups. This analysis revealed a significant decrease in SUV at week 6 among patients receiving NLAB (−3.6 vs. −0.95, *p*: 0.039, one-tailed test). The evaluation of invasive fungal infections (IFIs) using FDG-PET-CT has proven to be a valuable diagnostic tool in various studies. A Dutch study demonstrated that FDG-PET-CT added value to the management of 74% of patients with IFI, detecting lesions beyond the scope of anatomy-based imaging methods in 48.6% of cases [32]. An Australian study involving 48 cases of IFI similarly highlighted the utility of FDG-PET-CT in assessing response to antifungal therapy. Compared to computed tomography (CT), FDG-PET-CT detected sites of IFI dissemination in 35% of patients, as opposed to just 5% with CT [33]. Kim et al. further confirmed the progressive resolution of hypermetabolic nodules following antifungal treatment in five patients tracked sequentially with FDG-PET-CT [12].

The study suggests an improved performance of serum IFI markers in patients who received NLAB. However, this conclusion lacks statistical confirmation. By the third week, five out of the seven patients in the placebo group still had positive serum GM, compared with only two out of six patients treated with NLAB. Despite FDA and EMA’s preference for Platelia *Aspergillus* Ag (Bio-Rad) as a reference for GM detection, this study utilized Ag Virclia Monotest (Vircell S.L.) due to its favorable correlation and greater convenience. One study demonstrated that both methods can automate the process, but Virclia offers the advantage of individual sample processing without the need for additional single-dose strips for controls. This showed a concordance of 288 for 327 analyzed samples (к = 0.722) [34].

A recent study from Barcelona showed a significant decrease in IPA in patients with SARS-CoV-2 (CAPA) who had undergone solid organ transplants following prophylactic treatment with NLAB [35]. Our team also recently verified NLAB’s beneficial effects in an outbreak of CAPA in the intensive care unit of our center [36]. Positive results have also been achieved with the use of nebulized amphotericin B lipid complex (ABLC) [37,38]. The findings of the MUCONAB trial, which compared intravenous liposomal amphotericin B alone (the control group, 3–5 mg/kg/day, n: 15) with the combination of intravenous liposomal amphotericin B and nebulized amphotericin B deoxycholate (NAB, 10 mg twice a day, every other day, n: 17) in patients with mucormycosis, have recently been published. While there was no significant difference in treatment success between the two groups, none of the patients discontinued treatment [39].

Liposomalization has been proven to be a safe and effective method for various antimicrobials, as it increases drug concentration at infection sites while minimizing pharmaceutical toxicity. Future possibilities might involve the use of inhalable amphotericin B proliposomal microparticles formulated with lung-surfactant-like phospholipids. They can easily be administered using an FDA-approved dry powder device named Handihaler^®^ (Boehringer Ingelheim, Ingelheim, Germany), making it practical for clinical settings [40].

In essence, NAIFI is the inaugural randomized, controlled clinical trial to examine the efficacy of liposomal amphotericin B as a supplemental treatment for IPA. A significant limitation is its small sample size; however, this phase I-IIa pilot study was intended to test this treatment’s safety, which has been effectively affirmed by the study. Originally, the design of the study was blind, but the Spanish drug agency (AEMPS) did not approve the use of dyes for placebo masking. The lack of prior trials restricted the determination of optimal treatment dosage, duration, and monitoring times using FDG-PET-CT. The study largely included onco-hematological patients with angio-invasive IPA forms, suggesting that nebulized therapies may be more effective in broncho-pulmonary forms. ICU patients were not included due to initial concerns about the potential risks of amphotericin B precipitating in breathing tubes, although recent research has confirmed this to be a safe procedure. Despite these shortcomings, this study verifies the good tolerance and safety of NLAB. The indirect effectiveness data gathered from image monitoring (FDG-PET-CT) and biomarkers are promising, warranting further, larger-scale multicenter studies.

## Figures and Tables

**Figure 1 jof-10-00191-f001:**
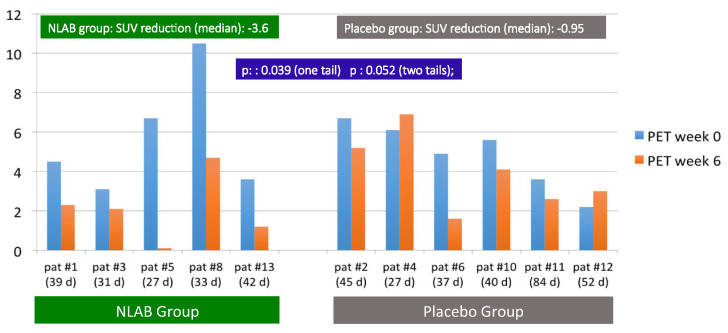
F2G/PET/CT uptake. SUV at inclusion (week 0) and at the follow-up (week 6). Patients in both groups and in real time (days) between two procedures.

**Figure 2 jof-10-00191-f002:**
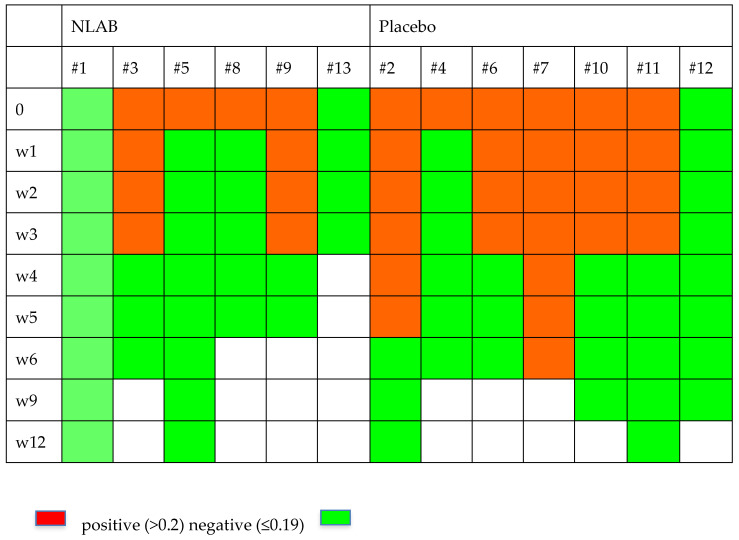
The serum galactomannan outcomes in both groups.

**Figure 3 jof-10-00191-f003:**
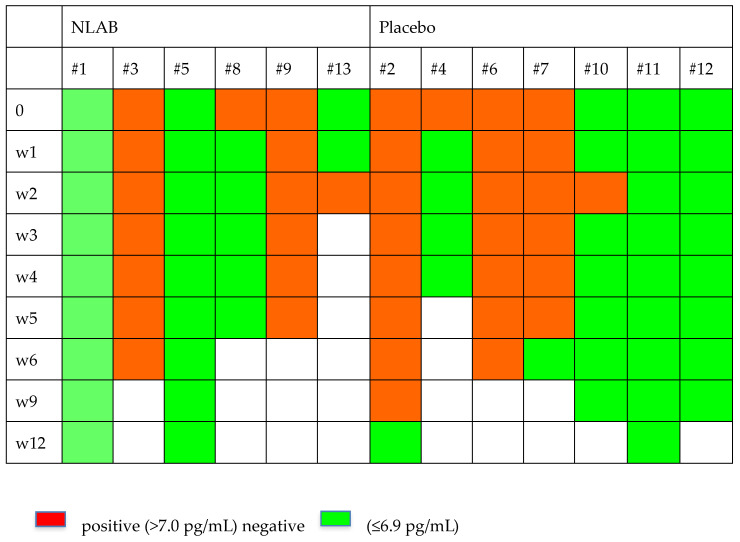
The serum B-D-Glucan outcomes in both groups.

**Table 1 jof-10-00191-t001:** Baseline clinical characteristics, randomization, and outcome of patients.

Patient	NLAB/ Placebo	<100 WBC/μL	Basal Condition	*Aspergillus* Species	SyAFT at Day 0, SyAFT Duration Days	Outcome (Week +12)	
						Dead	Global Response
#1	NLAB	No	Rheumatoid arthritis	AFC	Voriconazole, 44 d	No	Complete
#2	Placebo	Yes	Acute myeloid leukemia, IT	*A. terreus*	Voriconazole, 72 d	No	Partial
#3	NLAB	Yes	Acute lymphoblastic leukemia, IT	AFC	Voriconazole, 31 d	Yes	Dead *
#4	Placebo	No	Malnutrition, alcoholism	AFC	Isavuconazole, 43 d	No	Partial
#5	NLAB	Yes	Acute myeloid leukemia, IT	AFC	Voriconazole, 80 d	No	Complete
#6	Placebo	No	Autoimmune hepatitis	AFC	IVLAB *, 84 d	No	Complete
#7	Placebo	Yes	Prolymphocytic T leukemia, AlT	AFC	Isavuconazole, >84 d	No	Not clinically evaluable
#8	NLAB	Yes	Multiple myeloma, AuT	AFC	Isavuconazole, 40 d	Yes	Dead *
#9	NLAB	No	Autoimmune hepatitis	AFC	IVLAB, 34 d	Yes	Not clinically evaluable
#10	Placebo	Yes	Acute lymphoblastic leukemia, AlT	*A. flavus*	Isavuconazole, 74 d	Yes	Dead *
#11	Placebo	No	Systemic pulmonary sclerosis	AFC	Isavuconazole, >84 d	No	No response *
#12	NLAB	Yes	Acute myeloid leukemia, IT	AFC	Isavuconazole, 70 d	No	Complete
#13	NLAB	Yes	Acute myeloid leukemia, IT	*A. terreus*	Voriconazole, 61 d	No	Partial

NLAB: nebulized liposomal amphotericin B; IVLAB: Intravenous Liposomal Amphotericin B; WBC: white blood cells; SyAFT: systemic antifungal therapy; IT: induction therapy; AlT: allogeneic transplant; AuT: autologous transplant, MM; AFC: *Aspergillus fumigatus* complex; * #3. The patient presented progression of her baseline disease (T leukemia) and died at week +9; the PET-CT at week +6 had shown an SUV reduction. * #6. The patient received intravenous liposomal amphotericin (three times/week) until liver transplantation (59 days); after transplantation, he finished treatment with isavuconazole (25 days). #7. The patient was admitted to the ICU at week +3, and nebulization was suspended. The 2nd PET-CT could not be performed. The patient remained at the end of follow-up (day +84) with isavuconazole with an image of paramediastinal lung abscess that finally required surgical resection (day +98). * #8. At week +5, the patient presented progression of baseline disease (metastatic MM); PET-CT scans advanced to week +5, confirming a significant reduction in SUV. The patient died on day 69. #9. On day +4, the patient was admitted to the ICU, required mechanical ventilation, and nebulized therapy was discontinued. Respiratory failure progressed and required ECMO. The patient died in the ICU on day +34. No PET-CT scan was performed at week +6. * #10. Treatment with isavuconazole was maintained. She presented a relapse of CLL in the CNS. In week +12, she developed COVID-19 pneumonia, and BAL continued to exhibit *A. flavus* (GM-positive, 0.2). She was switched to IV liposomal amphotericin B. The patient died on day +74 due to respiratory failure and septic shock. * #11. The patient was treated with systemic corticosteroids. Isavuconazole was discontinued in week +6, but in week +9, it was reintroduced due to increased cough, expectoration, the persistence of *Aspergillus* in the respiratory specimen and the cavitated image. After negative cultures at week +15, the patient was maintained with nebulized amphotericin B. #13. In week +6, the medullary recurrence of the underlying disease was confirmed (80% blasts). The patient died after a follow-up on day 106.

**Table 2 jof-10-00191-t002:** Percentage of variation in post-nebulization FEV1 with respect to pre-nebulization FEV1.

Patient	NLAB/ Placebo	Week 1	Week 2	Week 3	Week 4	Week 5	Week 6	Mean
		Ratio (%)	Ratio (%)	Ratio (%)	Ratio (%)	Ratio (%)	Ratio (%)	Ratio (%)
#1	NLAB	−11	**−20**	−12	**−21**	**−21**	−7	−14
#2	Placebo	−5	−10	−17	−17		−15	−13
#3	NLAB	+11	−15	−11	−2	+32		+3
#4	Placebo	−2	−6	−4				−4
#5	NLAB	+1	−1	−2	−2	−3		−2
#6	Placebo	−2	−5	−14	**−25**	**−31**	−1	−13
#7	Placebo							-
#8	NLAB	−9	0	+14				+5
#9	NLAB							-
#10	Placebo	+34	−1	−4	−4	−8	−2	+3
#11	Placebo	−1	0	−1	−3	0	+1	−1
#12	NLAB	−2	−5	−4	+1	+5	+5	0
#13	NLAB	−9	−10	−9				−9

No significant decreases in FEV1 were observed between patients with NLAB or placebo nebulization. Reductions greater than 20% pre-specified as significant were only observed in three determinations of the same patient with NLAB (#1) and in two determinations of a patient receiving the placebo (#6).

**Table 3 jof-10-00191-t003:** Percentage of variation in post-nebulization FVC with respect to pre-nebulization FVC.

Patient	NLAB/ Placebo	Week 1	Week 2	Week 3	Week 4	Week 5	Week 6	Mean
		Ratio (%)	Ratio (%)	Ratio (%)	Ratio (%)	Ratio (%)	Ratio (%)	Ratio (%)
#1	NLAB	−16	−4	−9	**−27**	−21	−12	−15
#2	Placebo	−1	−7	−6	−12		−2	−6
#3	NLAB	+19	−15	−10	−2	−10	−2	−3
#4	Placebo	−2	−2	−26				−10
#5	NLAB	+6	−3	0	+1	0	−1	+1
#6	Placebo		−1	−4	−27	−25	−1	−2
#7	Placebo							-
#8	NLAB	−5	−1	−2				−3
#9	NLAB							-
#10	placebo	+31	−4	0	−4	−5	−3	+2
#11	placebo	−2	0	−2	−9	0	−1	−4
#12	NLAB	−11	−1	−8	−6	+8	−7	−4
#13	NLAB	−9	−21	−13				−14

No significant decreases in FVC were observed between patients with NLAB or placebo nebulization. Reductions greater than 20% pre-specified as significant were only observed in two determinations of the same patient with NLAB (#1) and in two determinations and one determination of two patients receiving placebo (#6, #13).

## Data Availability

Data are contained within the article.

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
