# Peer review of "A Phase I/IIa Prospective, Randomized, Open-Label Study on the Safety and Efficacy of Nebulized Liposomal Amphotericin for Invasive Pulmonary Aspergillosis"

_jof, 2024, doi:10.3390/jof10030191_

Round 1

Reviewer 1 Report

Congratulations on a very well conceived and performed randomized trial. You show that nebulized liposomal amphotericin B (NLAB) can be safely administered to hematological patients with invasive aspergillosis, and your data suggest that it perhaps could improve the response to standard treatment. The excellent outcome in your patients, unfortunately, implies that a much larger sample will be necessary to show clinical benefit.

I have some questions regarding treatment. Was therapeutic drug monitoring used for patients on voriconazole? Did any patient receive any echinocandin?

Given that you spend so much of your discussion on the potential non-statistically significant clinical benefits of NLAB I think you should include some succint comment regarding the mechanism why "topical" therapy (since there is no amphotericin detected in serum) is still expected to be effective in angioinvasive fungal infection (e.g., something to the effect that "the intraalveolar concentration of LAB is so-and-so and XXX above the common MIC, suggesting that a concentration gradient may result on delivery of effective amounts"). Or something like that. Efficacy in prophylaxis is easier to understand.

These are trivial: 

Page 2, line 70 and page 12, line 346, 348, 351 and 353: "preventive" is the preferred adjective, although "preventative" is acceptable (according to the Oxford Dictionary).

Figure 1: "tail", not "tale" on the blue box.

Author Response

Thank you very much for your comments.

Regarding therapeutic drug monitoring used for patients on voriconazole this was carried out in the 5 patients who received it and the dose was adjusted to maintain serum levels between 1-5 ug/ml.

The following text has been included in the results: therapeutic drug monitoring was carried out in patients receiving voriconazole and the dose was adjusted to maintain serum levels between 1-5 ug/ml. No TDM was performed in patients receiving isavuconazole

We agree to include in the discussion the importance of achieving high alveolar amphotericin B concentrations since there was no amphotericin B detected in serum. The following text has been added to the discussion:

Despite the angio-invasive nature of aspergillosis in the hematological patient, the importance of achieving high alveolar concentrations of amphotericin B in these patients is crucial for infection control since amphotericin B is not detected in the serum. Monforte et al (3) they demonstrated that amphotericin B concentration is high on the firs day (>11 ug/ml) after the administration of a 25 mg dose of n-LAB and remains elevated one week later (>4 ug/ml) above the common MIC, suggesting that a concentration gradient may result on delivery of effective amounts

Figure 1 has modified including tails (no tales)

Reviewer 2 Report

The study presented by Fortun et al is the first randomized study on the safety of NLAB use as therapy for patients with invasive pulmonary aspergillosis (IPA). The study presents a too small sample size to be conclusive.

Although the study presented by Fortun et al is not conclusive due to its small sample size, as the authors point out, it is a valid pilot study, with interesting results that could present the basis for a more robust study, to determine the use of NLAB as a valid therapy, instead of the conventional treatment with Amphotericin B for IPA. I have no further comments for the authors. 

Author Response

Thank you very much for your comments.
